# Synthesis and Biological Evaluation of a Caffeic Acid Phenethyl Ester Derivatives as Anti-Hepatocellular Carcinoma Agents via Inhibition of Mitochondrial Respiration and Disruption of Cellular Metabolism

**DOI:** 10.3390/cancers18010092

**Published:** 2025-12-27

**Authors:** Hao Dong, Yuan Gao, Dongyue Jiang, Chenjie Feng, Xinyue Gu, Xiyunyi Cai, Yulin Liu, Guangyu Zhang, Jiacheng Wen, Weiwei Diao, Ying Zhou, Ruixin Li, Dayang Xu, Weijia Xie, Liang Wu

**Affiliations:** 1State Key Laboratory of Natural Medicines, China Pharmaceutical University, Nanjing 210009, China; dongh-cpu@outlook.com (H.D.);; 2Department of Biology, The Johns Hopkins University, Baltimore, MD 21218, USA; 3School of Life Science and Technology, China Pharmaceutical University, Nanjing 210009, China; 4Department of Medicinal Chemistry, School of Pharmacy, China Pharmaceutical University, Nanjing 210009, China; 5School of Traditional Chinese Pharmacy, China Pharmaceutical University, Nanjing 210009, China

**Keywords:** hepatocellular carcinoma, ferroptosis, mitochondrial, energy metabolism, NAD^+^, NDUFS2

## Abstract

Hepatocellular carcinoma (HCC) is characterized by high incidence and mortality rates, yet effective therapeutic drugs remain scarce, creating an urgent need for viable treatment strategies. This study aimed to develop novel small-molecule anti-HCC agents based on the naturally derived antitumor compound caffeic acid phenethyl ester (CAPE), evaluate their efficacy against HCC, and investigate their underlying pharmacological mechanisms. A total of 28 structurally analogous small-molecule derivatives were synthesized, from which the most potent compound, designated WX006, was selected. Using an integrated approach encompassing molecular biology, cell biology, computational biology, and multi-omics analysis, we demonstrated that WX006 effectively depletes Nicotinamide adenine dinucleotide (NAD^+^) in HCC cells by binding to NADH: Ubiquinone Oxidoreductase Core Subunit S2 (NDUFS2), thereby disrupting mitochondrial function and intracellular metal ion homeostasis. This study provides valuable and independent insights into the development of derivatives based on the natural small molecule CAPE and establishes a further foundation for the advancement of anti-HCC therapies.

## 1. Introduction

Hepatocellular carcinoma (HCC) is the most common type of primary liver malignancy [1,2]. According to the Global Cancer Observatory, HCC accounted for over 800,000 new cases and around 700,000 deaths in 2020, making it the sixth most common cancer and the third leading cause of cancer death globally [3]. The rising incidence of HCC is closely associated with an increased prevalence of cirrhosis, primarily due to chronic infections with hepatitis B and C, alcoholic liver disease, and non-alcoholic fatty liver disease [4].

The therapeutic landscape for HCC has evolved significantly over the past decade. While surgical resection remains the primary treatment option for patients with early-stage HCC, many patients with non-resectable tumors or additional complications still await effective and feasible interventions [5]. In addition, many patients with HCC who undergo surgical resection still face the risk of cancer recurrence [6]. To address these gaps in treatment outcomes, recent advancements have introduced systemic therapies, including multi-kinase inhibitors such as sorafenib and lenvatinib, to provide modest survival benefits [7,8]. Additionally, immune checkpoint inhibitors such as nivolumab and pembrolizumab have emerged as promising options for patients with HCC resistant to standard treatments [9,10,11].

A thorough understanding of the metabolic characteristics of HCC is essential for developing targeted therapies. HCC pathogenesis is driven by profound metabolic rewiring of neoplastic cells. The mitochondrial electron transport chain (ETC) plays a crucial role in cellular respiration and ATP production via oxidative phosphorylation [12]. While glycolysis enables quick energy production for cancer cells’ growing metabolic demands [13], mitochondria remain an indispensable component for cancer cell homeostasis not only in energy production but also in biosynthetic metabolism and regulation of cell signaling [14]. Impaired oxidative phosphorylation not only depletes ATP reserves but also generates cytotoxic ROS surges that activate caspase-dependent apoptosis [15]. For example, recent studies show that inhibiting Complex I of the ETC impairs ATP synthesis and triggers cell death across various cancer types [16,17,18]. However, challenges remain in clinical translation of Complex I inhibitors due to potential off-target side effects.

The interplay between metal ions and regulated cell death mechanisms, particularly ferroptosis and cuproptosis, has gained traction in recent research [19]. Ferroptosis—an iron-dependent cell death modality driven by phospholipid peroxidation [20,21], and cuproptosis—a copper-mediated process involving mitochondrial dysfunction [22], represent mechanistically distinct yet therapeutically synergistic pathways for targeting the metabolic vulnerabilities of HCC [23]. Both forms of cell death involve cancer cells’ efficient metabolic adaptations and stress responses for survival. Given that metabolic dysregulation and oxidative stress are key features of the HCC microenvironment, targeting both ferroptosis and cuproptosis has been highlighted as a novel therapeutic avenue for HCC treatment [24,25,26]. Inducing ferroptosis has been shown to sensitize HCC cells to various treatments [27], and manipulating copper levels to promote cuproptosis can further diminish tumor viability [28].

Caffeic acid phenethyl ester (CAPE, Figure 1), a key bioactive constituent of propolis, has been established to possess a range of significant biological properties [29], including antioxidant [30,31], anti-inflammatory [32], and neuroprotective effects. As a well-established broad-spectrum antitumor agent, CAPE demonstrates inhibitory effects against a range of cancers both in vitro and in vivo, including hepatocellular carcinoma [33,34], glioma [35,36], melanoma [37], lung cancer [38] and prostate carcinoma [39].

Owing to its significant inhibitory effect on the NF-κB signaling pathway, CAPE is frequently employed as a canonical NF-κB inhibitor in numerous studies [40]. However, its biological functions extend beyond this role. CAPE has also been reported to suppress tumor growth and metastasis by modulating multiple signaling pathways, including the inhibition of matrix metalloproteinases (MMPs), suppression of p53 ubiquitination and degradation [41], and interference with the PI3K/AKT/XIAP axis [37].

Furthermore, CAPE can act synergistically with conventional chemotherapeutic agents or other therapies, enhancing their efficacy against specific malignancies, as demonstrated by studies from L. K. Kuo et al. [38] and Y. K. Fu et al. [42]. In contrast to its direct antitumor effects, the synergistic mechanism of CAPE primarily functions through auxiliary pathways, such as inducing DNA damage response [43] and impairing proteasome function [44].

CAPE demonstrates a favorable safety profile and a low incidence of adverse effects in the contexts of both its direct antitumor activity and its chemosensitizing effects. A study in 1988 already established that CAPE can effectively inhibit abnormal cell proliferation at a concentration of 2 μg/mL. Importantly, even at a five-fold higher concentration (10 μg/mL), it displayed no cytotoxic effects on normal murine cells [45]. Its pronounced antitumor and immunomodulatory activities further support its development as a lead compound for novel small-molecule anticancer agents or as a core structure for derivative design.

Recent years have witnessed considerable progress in the development of CAPE derivatives [46,47]. The structural features of CAPE render it a promising scaffold for chemical modification (Figure 2). Consequently, researchers have synthesized a diverse array of CAPE derivatives, many of which exhibit pharmaceutical value comparable to, or even surpassing, that of the parent compound. These derivatives have demonstrated notable efficacy in various biological activities, including antitumor [48] and neuroprotective [49] effects. Most retain the antioxidant and anti-inflammatory [50] activities inherent to CAPE, while also largely inheriting its favorable safety profile.

As summarized in Figure 3, we have collated and analyzed various existing modification strategies for CAPE derivatives with antitumor activity. Current research indicates that most of these derivatives are chemically modified at three key sites: the hydrolytically labile ester bond, the catechol moiety (o-dihydroxy group), and the phenethyl moiety, with such structural optimizations demonstrating considerable antitumor potential [39,48,51,52,53,54].

These results, consistent with subsequent studies, underscore CAPE’s high biocompatibility, selective cytotoxicity toward tumor cells, and wide therapeutic window. Based on these characteristics, CAPE was selected for the development of a new generation of antitumor small molecules. This study synthesized a series of CAPE derivatives and investigated their antitumor pharmacological mechanisms against hepatocellular carcinoma. The findings enhance our understanding of the structure-activity relationships of these derivatives and provide new insights for optimizing their physicochemical properties and biological activities.

## 2. Experimental Section

### 2.1. Chemistry Methods and Data

All experimental materials, methods and details of the experiments can be referred to in the Appendix A.

### 2.2. Biological Materials and Methods

#### 2.2.1. Sources of Inhibitors, Compound, Supplements

Z-VAD-FMK (MCE, Shanghai, China, HY-16658B), Necrostatin (MCE, HY-15760), VX765 (MCE, HY-13205), DIDS sodium salt (MCE, HY-D0086), Ferrostatin-1, (MCE, HY-100579), Deferoxamine mesylate (MCE, HY-B0988), ISRIB (MCE, HY-12495), 4-Phenylbutyric acid (4-PBA), (MCE, HY-A0281), Ammonium tetrathiomolybdate (ATTM), (Aladdin, Shanghai, China, A189030-200mg), BAPTA-AM (Aladdin, B115502-25mg), 2-Aminoethyl Diphenylborinate (2-AD) (Psaitong.Inc., Beijing, China, A10085-1g), Trolox (MCE, HY-101445), Mito-TEMPO (MCE, HY-112879), N-Acetyl-L-cysteine (NAC) (Beyotime, Shanghai, China, ST2524-5g), L-Glutathione reduced (GSH), (Beyotime, ST2474), Sorafenib (Aladdin, S125098), Disodium Bathocuproinedisulfonate (Aladdin, D303893-100mg), Mdivi-1 (MCE, HY-15886), nicotinic acid (Yuanye, Shanghai, China, S13023), NAD^+^ (MCE, HY-B0445), succinic acid (Psaitong.Inc., China, S10107), malic acid (Psaitong.Inc., China, M70012),were obtained from the indicated vendors.

#### 2.2.2. Cell Line, Culture Procedures and Treatment

Huh7 and Hep3B cell lines were purchased from the Shanghai Cell Bank of the Chinese Academy of Sciences, while the H22 mouse ascites tumor cell line was obtained from the Wuhan Cell Bank of the Chinese Academy of Sciences. The Hepa1-6-Luc cell line was purchased from iCell Bioscience Inc. (Shanghai, China). All cell lines were authenticated using STR profiling and tested for mycoplasma contamination prior to use. All cells were grown in an incubator at 5% CO_2_ and 37 °C. For in vitro culture, the cells were maintained in 10% FBS (PAN Premium) and 10% DMEM (KGM12800H from Kaiji (Nanjing, China), supplemented with 1% penicillin/streptomycin) during the logarithmic phase of growth.

#### 2.2.3. Animal Husbandry and Ethical Approval

The animal study protocol was approved by the ethics committee of China Pharmaceutical University. This study was conducted in accordance with the Guidelines of the Care and Use of Laboratory Animals issued by the Chinese Council on Animal Research.

SPF-grade Kunming mice, male, aged 3 weeks, were purchased from Hangzhou Ziyuan Experimental Animal Technology Co., Ltd. (Hangzhou, China), with license number SYXK (Su) 2018–2019. The Kunming mice were divided into groups of five and housed in a conventional animal research facility, where they were provided with purified water and experimental feed. The environmental temperature was maintained between 20 and 26 °C, with humidity controlled between 40 and 70%. A regular light/dark cycle was maintained, and the food and bedding were changed every three days.

SPF-grade C57BL/6 mice, male, aged 6 to 8 weeks, weighing 19 to 21 g, were purchased from Hangzhou Ziyuan Experimental Animal Technology Co., Ltd., with license number SYXK (Su) 2018–2019. The mice received SPF mouse chow and were allowed to drink sterile water ad libitum for 7 days before the experiments.

The C57BL/6 mice were housed in groups of five, with the environmental temperature maintained between 20 and 26 °C and humidity controlled between 40 and 70%. The light/dark cycle was consistent, and the mice had free access to water and food, with changes to the water and bedding occurring every three days.

#### 2.2.4. Culture of H22 Mouse Ascites Tumor Cells and Establishment of Ectopic Tumor Models

The detailed protocols for these experiments are available in the Appendix A.

#### 2.2.5. Establishment of the Hepa1-6-Luc In Situ HCC Model and Fluorescent In Vivo Imaging

The detailed protocols for these experiments are available in the Appendix A.

#### 2.2.6. Colony Formation Assay

Huh7 and Hep3B cells were plated at a density of 500 cells per well in a 6-well cell culture plate, and the medium was supplemented to a final volume of 2 mL. The plates were incubated at 37 °C in a 5% CO_2_ atmosphere for approximately 24 h. Once the cells adhered, the culture medium was discarded, and each well was treated with either blank medium or medium containing WX006 (0, 5, and 10 μM). The cells were then incubated at 37 °C in a 5% CO_2_ atmosphere with the drug for 48 h.

The medium was replaced every 2–3 days. When visible cell colonies formed on the bottom of the plate, the culture medium was removed, and the wells were washed once with pre-cooled PBS. Subsequently, 4% paraformaldehyde was added to fix the cells at room temperature for 15 min, followed by air-drying. Each well received 1 mL of freshly prepared 1× Giemsa stain working solution and was stained in the dark for 15–30 min. After washing with distilled water, the wells were allowed to air dry, and photographs were taken to count the number of cell colonies.

#### 2.2.7. Scratch Assay

When the Huh7 and Hep3B cells reached a stable growth state with a confluence of 80% to 90%, the cells were trypsinized, centrifuged, and counted. Approximately 300,000 cells were added to each well of a 12-well plate and mixed gently. The culture was continued until the cells adhered and covered the bottom of the plate. The culture medium was then discarded, and the lid of the 12-well plate was placed in the central position above the wells. Using a 200 μL pipette tip, a straight line was drawn along the edge of the lid touching the bottom of the plate. The wells were then washed twice with 1 mL of PBS to remove any cells disrupted during the marking process.

Subsequently, 1 mL of fresh serum-free medium or serum-free medium supplemented with WX006 was added to each well. The 12-well plate was placed under an inverted microscope for observation and imaging. Afterward, the plates were returned to the incubator and further cultured for an additional 24 and 48 h, with photographs taken at each time point. The images were processed using Photoshop 2024 and ImageJ software (version:1.54i), and the migration rate was calculated.

#### 2.2.8. Western Blot

Cells were lysed using RIPA buffer (Beyotime P0013B). Total protein was quantified using a BCA Protein Quantification Kit (Vazyme, Nanjing, China, E112-02). Samples were subjected to SDS–PAGE 10%/12% (Vazyme E303/E304) and transferred to a PVDF membrane (Bio-Rad, Hercules, CA, USA, Cat 1620177). The primary antibodies used in this study were: rabbit anti-NDUFS2 (Proteintech, Wuhan, China 28125-1-AP, 1:2000), Secondary antibodies: Anti-Rabbit IgG, HRP-linked Antibody (1:10,000; CST, Boston, MA, USA, 7074), the blots were then scanned with ChemiDoc Imaging System (Bio-rad, USA) and quantified using ImageJ software.

#### 2.2.9. Cell Proliferation Assays

Huh7 and Hep3B cells were seeded in a 96-well plate (Corning, New York, NY, USA 3599) at a density of 3000 cells per 100 μL. After 24 h of incubation, half of the medium was replaced with fresh complete medium containing the drug or drug combination. After WX006 treatment or combination treatment, cell proliferation rate was measured using CCK-8 reagent (Apex Bio, Houston, TX, USA, K1018). Briefly, CCK-8 reagent was added into each well and incubated for 4 h. The absorbance at 450 nm was measured with a microplate reader (TECAN, Zurich, Switzerland). Each combination orthogonal group or single treatment detection group included at least three replicates.

#### 2.2.10. Preparation Methods for Drugs or Drug Compositions

The detailed protocols for these experiments are available in the Appendix A.

#### 2.2.11. Transcriptome Sequencing and Library Construction Methods

Total RNA was isolated using the Trizol Reagent (Invitrogen Life Technologies, Carlsbad, CA, USA), after which the concentration, quality and integrity were determined using a NanoDrop spectrophotometer (Thermo Scientific, Carlsbad, CA, USA). The sequencing library was then sequenced on BGI-T7 (Hep3B) and BGISEQ-500 (Huh7) platform (BGI) Shanghai Personal Biotechnology Co., Ltd., Shanghai, China.

The detailed protocols for these experiments are available in the Appendix A.

#### 2.2.12. Non-Targeted Metabolomics Analysis

Untargeted metabolomics was conducted at BioNovoGene Co., Ltd. (Suzhou, China). The detailed protocols for this experiments are available in the Appendix A.

#### 2.2.13. Targeted Metabolomics Analysis

Huh7 cells (1 × 10^7^/sample) were treated with Vehicle or WX006 (10 μM) for 6 h and intracellular metabolites were analyzed by SRM/MRM technology using ultra-high-performance liquid-chromatography mass spectrometry (Agilent 1290 Infinity LC, Santa Clara, CA, USA). 1 mL extraction liquid (Vmethanol:Vacetonitrile:Vwater = 2:2:1) was added to the sample and vortexed for 60 s, followed by the low-temperature ultrasound twice for 60 min. The samples were left for 1 h to precipitate proteins and then the supernatant was obtained by centrifugation (14,000 rpm × 4 °C × 20 min). The supernatant was filtered by 0.22 μm membrane and transferred into the detection bottle for LC-MS detection (PANOMIX Biomedical, Suzhou, China)

#### 2.2.14. Electron Microscopy Imaging

After treatment with WX006, the tumor cells were washed three times with PBS and digested using 0.25% enzyme without EDTA. Following cell collection, the cells were washed three times with pre-cooled PBS. The samples were then prefixed with 2.5% glutaraldehyde and subsequently fixed with 1% osmium tetroxide.

Each copper grid was first observed under low magnification (8000×) to examine the entire tissue, followed by capturing images of specific areas of interest to observe the specific lesions (25,000×).

The detailed protocols for this experiment are available in the Appendix A.

#### 2.2.15. Mitochondrial Respiratory Complex (MRC) Activity Assay

Mitochondria from Huh7 cells were extracted using a mitochondrial isolation kit according to the manufacturer’s standard protocol. A Bradford Protein Assay Kit (Beyotime, Shanghai, China) was used to quantify the mitochondrial samples. The activity of the MRC complexes was assessed by colorimetry using commercial kits (Abbkine, KTB1850, KTB1860, KTB1870, KTB1880, Wuhan, China) with a microplate reader (TECAN, Swiss) following manufacturer instructions.

#### 2.2.16. RNA Isolation and Quantitative PCR (qPCR)

RNA was isolated using the FastPure Cell/Tissue Total RNA Isolation Kit (Vazyme, RC101) according to the manufacturer’s instruction. Reverse transcription (RT) was performed using oligo-dT primers (Vazyme, R223). qPCR analysis was performed using SYBR qPCR Master mix (Vazyme, Q312) and were performed on the CFX96 qPCR System (Bio-Rad, USA). RNA was normalized to Gapdh expression and calculated as delta-delta threshold cycle (ΔΔC_t_). The primers used for qPCR are summarized in Appendix A.

#### 2.2.17. Detection of Various Biochemical Data Using Assay Kits

Cell glutathione (GSH) and GSSG content were determined by a commercial kit (Beyotime, S0053, China) according to the manufacturer’s protocol. Cell NAD^+^ and NADH content were determined by a commercial kit (Beyotime, S0175, China) according to the manufacturer’s protocol. Cell ATP content were determined by a commercial kit (Beyotime, S0026, China) according to the manufacturer’s protocol. Cell NADP^+^ and NADPH content were determined by a commercial kit (Elabscience, E-BC-K803-M, Wuhan, Hubei, China) according to the manufacturer’s protocol. Cell MDA content were determined by a commercial kit (Beyotime, S0026, China) according to the manufacturer’s protocol. Cell Fe^2+^ content were determined by a commercial kit (Elabscience, E-BC-K881-M, China) according to the manufacturer’s protocol. Cell Cu^2+^ content were determined by a commercial kit (Elabscience, E-BC-K775-M, China) according to the manufacturer’s protocol.

#### 2.2.18. Fluorescent Imaging and Staining

Commercially prepared cell culture slides (BS-14-RC, Biosharp, Hefei, Anhui, China) were placed in a 12-well plate under sterile conditions. HCC cells in the logarithmic growth phase were adjusted to a cell density of 5 × 10^4^ to 1 × 10^5^ cells/mL using complete culture medium. The plate was gently shaken to ensure uniform distribution of the cells, avoiding displacement of the slides. The plate was then incubated at 37 °C in a 5% CO_2_ incubator for 24 h.

#### 2.2.19. Iron Staining

5 × 10^4^ Huh7 and Hep3B cells were seeded on Nest, 35mM glass bottom culture dishes. As above, cells were pre-incubated for 24 h with or without different concentration of WX006. Cells were grown for an additional 24 h. Plates were washed 3× in HBSS and stimulated at 37 °C, 21% oxygen and 5% CO_2_ for 20 min in HBSS. Next cells were stained in 1 μM Ferro orange (dojingo) in HBSS for exactly 30 min at 37 °C, 21% oxygen and 5% CO_2_ and imaged immediately. Treatments were staggered to ensure precise staining duration. Images were acquired with immunofluorescence microscopy (IX73, Olympus, Tokyo, Japan). Five representative fields were captured for each condition under identical exposure times.

#### 2.2.20. Mitochondrial Membrane Potential (MMP) Assay

The MMP of Huh7 and Hep3B cells was determined using JC-1 probe to proof the WX006 mediated variation in MMP as an indicator of mitochondrial dysfunction. To detect change in MMP, Huh7 and Hep3B cells were transferred to 6 well plate at density of 5 × 10^5^ cells per well and cultured for 24 h at 37 °C. The used media was discarded and the cells were treated with 2 mL culture medium containing different dose of WX006 for 6h at 37 °C. Untreated cells served as control group, while cells having only culture medium without any staining were used as blank. After treatment for 6 h, the cells were stained with JC-1 according to the manufacturer’s protocol. Images were acquired with immunofluorescence microscopy (IX73, Olympus, Japan). Five representative fields were captured for each condition under identical exposure times.

#### 2.2.21. Quantification of Intracellular Calcium Ion Levels

The intracellular calcium ion levels of Huh7 and Hep3B were measured with the diluted Fluo-4 AM ester stock solution (Beyotime, China) (4 μM with PBS). After culturing with WX006 (10 μM) for 6 h, the culture medium was removed, and the cells were rinsed with PBS for three times. The Fluo-4 AM ester solution was added (500 μL per well) and incubated for 30 min at 37 °C. Then, after removing the Fluo-4 AM ester solution, the cells were rinsed by PBS for three times. Cells were stained with DAPI (Beyotime, China) for 10 min at RT. After rinsing, the fluorescent images of Huh7 and Hep3B are captured by a immunofluorescence microscopy (IX73, Olympus, Japan).

#### 2.2.22. Cell Proliferation Assay

The EdU-488 kit (Beyotime, cat. no. C0071L) was used to assess the proliferation capability of tumor cells treated with WX006 following the manufacturer’s instructions.

#### 2.2.23. Reactive Oxygen Species (ROS) Detection

Dichlorofluorescein diacetate (DCFH-DA, Beyotime, China) was used to quantify the total intracellular ROS. Cells were washed three times with PBS and then cultured in medium supplemented with 10 μM DCFH-DA. The samples that had undergone treatment were placed in a cell growth incubator for a duration of 30 min, with protection from light. The fluorescence emitted by the cells was then examined using a immunofluorescence microscopy (IX73, Olympus, Japan) and documented with pictures. The fluorescence intensity was measured by image J software to evaluate the intracellular reactive oxygen species content.

#### 2.2.24. Immunofluorescent Staining

For IF staining, after treating Huh7 and Hep3B cell culture slides with different concentrations of WX006 for 6 h, the original medium was discarded. The slides were washed three times with ice-cold HBSS, followed by fixation with 4% pre-cooled paraformaldehyde for 15 min. After washing three times with HBSS, the slides were permeabilized with 0.5% Triton X-100 for 15 min. After three more washes, the slides were blocked with 5% BSA at room temperature for 2 h, anti-LC3 (1:200; Proteintech, Wuhan, Hubei, China, 14600-1-AP) antibody, anti-p62 (1:500; CST, 88588, Massachusetts, USA) antibody, anti-DLAT (1:200; CST, 2439) antibody, primary antibody were incubated overnight at 4 °C in a humidified atmosphere. After washing, sections and cells were incubated with anti-rabbit Alexa Fluor 488 (1:500; Abcam, Cambridge, UK, ab150073) secondary antibodies for 1.5 h at room temperature. Nuclear counterstaining was performed by DAPI. Slides were then imaged using an immunofluorescence microscopy (IX73, Olympus, Japan). For quantitative analysis, ImageJ (NIH) was used to count the staining intensity of fluorescent.

#### 2.2.25. Drug Affinity Responsive Target Stability (DARTs)

Huh7 and Hep3B cells protein were extracted by ice-cold RIPA buffer. The protein solutions were diluted by TNC buffer (50 mmol/L Tris-HCl, pH 8.0, 50 mmol/L NaCl, 10 mmol/L CaCl_2_) and treated with WX006 (0, 50, 100, 500 and 1000 μmol/L) at room temperature for 2 h. Then, the protein solutions were incubated with 0.1% pronase or not at room temperature for 15 min. Subsequently, loading buffer was added immediately and heated to prepare for Western bolt to analyze the level of NDUFS2.

#### 2.2.26. Cellular Thermal Shift Assay (CETSA)

Huh7 and Hep3B cells was treated with WX006 (10 μM) or 0.1% DMSO for 2 h and then lysed by ice-cold NP-40 buffer. The lysate was equally divided into 8 tubes and heated at gradually rising temperatures (from 42 to 67 °C, T-100, Bio-rad) for 3 min. After cooling, the lysate was centrifuged at 12,000 rpm for 10 min at 4 °C, and the supernatant was collected and added loading buffer for subsequent Western bolt analysis.

#### 2.2.27. Molecular Docking

The atomic coordinates of Cryo-EM structure of respiratory complex I were obtained from the Protein Data Bank. (PDB ID: 6ZTQ) and the structure of WX006 was generated from Indraw. The adjusted protein model was centered around the original ligand piericidin A (the target), encompassing all subunits within a 30 Å radius, including Chains A, B, C, D, H, I, J, P, Q, V, W, Z, a, q, and r. Both protein and ligands were preprocessed with ChimeraX (v1.8) and then molecular docking and binding affinity calculations were performed using the UCSF DOCK6 software (v6.12). After obtaining the optimal conformation through DOCK6 docking, we further validated the results using Auto Dock Vina (v1.2.5) under rigid docking parameters, followed by comprehensive interaction mode analysis. Protein-ligand interaction analysis was performed using the Protein-Ligand Interaction Profiler (PLIP-2021) and PyMol (Molecular Graphics System, Version 3.0.0.) to systematically characterize binding modes, including hydrogen bonding, hydrophobic contacts, π-π stacking, halogen bonding, and salt bridges.

#### 2.2.28. Molecular Dynamics (MD) Simulation

The molecular dynamics simulations were performed using the 6ZTQ structure obtained from the Protein Data Bank (PDB) database. The top-ranked small-molecule conformation from DOCK6 molecular docking was exported as a mol2 file for subsequent analyses. The protein topology was generated using the Amber99sb-ildn force field in GROMACS (2020.6), while the ligand topology was constructed using the GAFF force field via the Sobtop tool (refer to http://sobereva.com/soft/Sobtop/ Tian Lu, Sobtop, Version [dev5, accessed on 15 July 2024]). The combined system yielded the ligand-protein complex files (.gro, .itp, and .top). A dodecahedral simulation box was constructed using GROMACS, solvated in an aqueous solvent system, and neutralized with NaCl. Following energy minimization and equilibration under NVT and NPT ensembles, a 100 ns molecular dynamics (MD) simulation was conducted with trajectory snapshots saved every 10 ps. The resulting trajectories were subjected to least-squares fitting and root mean square deviation (RMSD) calculations to evaluate structural deviations of both the protein backbone (BDH binding pocket) and the ligand. Hydrogen bond occupancy between the protein and ligand was subsequently analyzed.

The binding free energy of the protein-ligand complex was calculated using the MM-PBSA method via the gmxtools suite: (https://github.com/Jerkwin/gmxtools/blob/master/gmx_mmpbsa/gmx_mmpbsa.bsh. version:2024.5, accessed on 15 December 2024). The software was executed in a bash environment (Gitbash), generating a time-dependent binding energy profile.

#### 2.2.29. Statistical Analysis

Animal Experiments were performed with five biological replicates, each with three technical replicates, other Experiments were performed with three biological replicates, each with three technical replicates, to ensure reproducibility and allow accurate statistical measurements. Statistical analysis was performed using the GraphPad Prism 9.5 (GraphPad Software, Boston, MA, USA). Student’s *t*-test was used for the comparison of the differences between the two experimental groups. Statistical differences were estimated by one-way ANOVA, with Dunnett’s post hoc tests or ordinary two-way ANOVA, whichever was appropriate. Results of animal experiments are expressed as mean ± standard deviation (SD). Results of other experiments are expressed as mean ± SEM. Differences with *p* < 0.05 were regarded as significant. Asterisks indicate significant *p* values in the graphs, denoted as * *p* < 0.05, ** *p* < 0.01,*** *p* < 0.001.

## 3. Result

### 3.1. Chemistry

#### 3.1.1. Synthesis of Compounds

The detailed protocols for these experiments are available in the Appendix A.

#### 3.1.2. General Synthesis Route (As Show as Figure 4)

Starting from compound **21**, demethylation with boron tribromide (BBr_3_) proceeded smoothly to afford intermediate **22** in 99% yield. Subsequent protection of the phenolic hydroxyl group in **22** with MEMCl provided the protected derivative **23** in 99% yield. Compound **23** was then subjected to a Heck coupling reaction with ethyl acrylate, achieving near-quantitative conversion to ester **24**. After simple workup, the crude product **24** was directly hydrolyzed with potassium hydroxide and acidified to give the key intermediate **25**. This intermediate **25** was further functionalized via two distinct pathways: esterification with various primary alcohols using EDCI as a coupling agent, or Mitsunobu reaction with chiral secondary alcohols employing triphenylphosphine (TPP), to generate a series of diester intermediates. Finally, removal of the MEM protecting group from these diesters yielded the corresponding final products.

**Scheme 4 cancers-18-00092-sch004:**
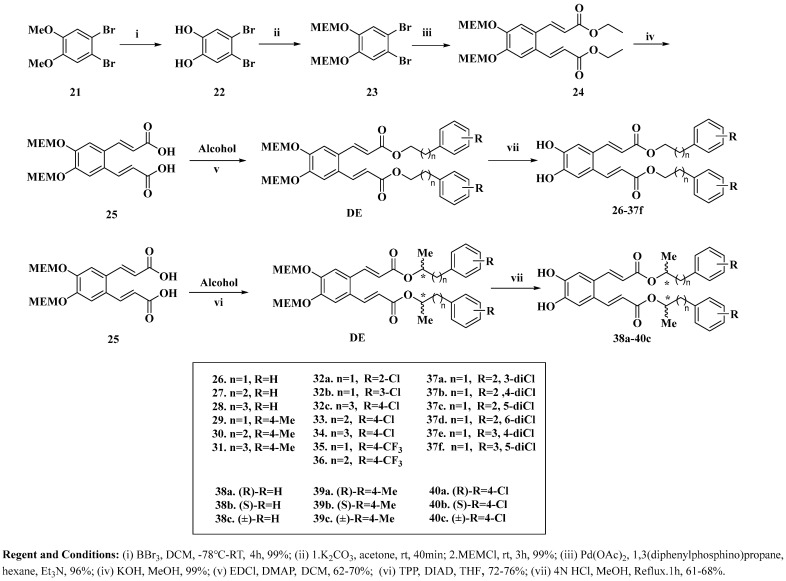
General Synthesis Route of WX006 and its analog.

### 3.2. Biology

#### 3.2.1. Evaluation of the Cytotoxic Activity of the Selected Compounds

Based on the IC_50_ values presented in Table 1, we derived preliminary structure–activity relationship (SAR) insights for this series of CAPE derivatives, which revealed several key structural determinants for antiproliferative activity. The α,β-unsaturated conjugated double bond was essential, as its removal significantly diminished activity. Retaining the free catechol structure was also crucial for antiproliferative efficacy. While the ester bond served as a core linkage, stereochemistry ubstantially modulated activity; for example, introduction of a methyl group at the α-position of the ester oxygen resulted in a pronounced stereochemical effect, with the R configuration exhibiting markedly superior activity to the S configuration. Modifications to the phenyl ring, whether with electron-donating or electron-withdrawing groups, did not significantly enhance antiproliferative activity, indicating that substituent electronic effects were not a key determinant in this series. In terms of side-chain optimization, extending the alkyl length up to C4 enhanced activity, whereas further extension (e.g., to C5) led to a decline. The most impactful strategy was the construction of a di-side-chain architecture, which conferred significantly stronger activity than mono-side-chain CAPE derivatives. Additionally, the length of the ester linker (n = 2–4) showed a positive correlation with anti-hepatocellular carcinoma activity. The high potency of WX006 corroborated these SAR insights: its di-side-chain design underlay its superior activity over CAPE; its activity aligned with the optimal ester linker length trend; and it retained the essential α,β-unsaturated double-bond pharmacophore. Notably, WX006 achieved maximal activity without relying on complex phenyl-ring substitutions, underscoring that optimization of this scaffold should focus on rational expansion of the overall structure and precise linker design rather than local aromatic substituent modifications.

#### 3.2.2. WX006 Suppresses HCC Cell Growth via Non-Apoptotic Mechanisms and Induces Ferroptosis Coupled with Cuproptosis, Effectively Impairing DNA Replication in HCC Cells

In our current research, our team discovered and structurally optimized a compound based on CAPE: WX006 (Figure 1A) (Diphenethyl 3,3′-(4,5-dihydroxy-1,2-phenylene) (2E,2′E)-diacrylate), and validated its antitumor growth effects across various HCC cell lines WX006 exhibited favorable growth inhibition rates in two representative HCC cell lines, Huh7 and Hep3B. The half-maximal inhibitory concentration (IC_50_) of WX006 against Hep3B cells was determined to be 7.485 μM at 24 h and 3.535 μM at 48 h. Similarly, the IC_50_ values for Huh7 cells were measured as 8.083 μM following 24 h exposure and 3.751 μM after 48 h treatment. (Figure 1B) Following treatment of hepatic primary cells isolated from C57 mice with WX006, cell proliferation rates were assessed via the CCK-8 assay. The results demonstrated minimal growth inhibition of WX006 on murine hepatic primary cells, indicating its low cytotoxicity toward normal cells and targeted inhibitory effects on tumor cells (Figure 1C). Furthermore, the colony formation assay demonstrated that WX006 effectively suppressed colony formation in Huh7 and Hep3B cells under in vitro conditions (Appendix A), while the wound healing assay revealed a significant inhibition of migration in both HCC cell lines (Appendix A).

Given the growth-inhibitory effects of WX006 on multiple HCC cell lines, our research team has undertaken an in-depth investigation into its antitumor mechanisms and sought to validate its pharmacological targets to assess its potential as a novel therapeutic agent for HCC. Flow cytometry analysis utilizing propidium iodide (PI) staining was performed to quantify DNA content in WX006-treated HCC cells, (Figure 1D,E) thereby assessing cell cycle alterations. The results demonstrated a significant reduction in the proportion of cells successfully entering the G2/M phase, accompanied by distinct G1/S phase arrest post-WX006 treatment. Flow cytometry analysis using Annexin V/PI (AV/PI) staining was employed to stage tumor cells at distinct apoptotic phases. (Figure 1F,G) The results demonstrated no significant increase in apoptosis-positive cells in Huh7 cells following WX006 treatment.

Notably, Hep3B cells exhibited distinct phenotypic characteristics compared to Huh7 after 24 h of WX006 exposure, with a marginal increase in apoptosis-positive cells observed at this time point. Combinatorial treatment of WX006 with inhibitors such as Z-VAD-FMK (apoptosis inhibitor) failed to rescue the inhibitory effects on cell viability (Figure 1H). The proliferative status of HCC cells was evaluated via EdU assay, which demonstrated a marked decrease in the proportion of EdU-positive cells (indicative of actively proliferating populations) following WX006 treatment (Figure 1I). Furthermore, DNA fiber analysis revealed pronounced replication fork stasis in HCC cells post-treatment, characterized by DNA fragmentation and substantial accumulation of single-stranded DNA (ssDNA) (Figure 1J). These findings collectively suggest the induction of replication stress, which likely underlies WX006-mediated cell cycle arrest in HCC cells. Building upon these findings, we further subjected HCC cells to alkaline comet assay to assess DNA damage. The results indicated that, compared to the control group, the WX006-treated group exhibited negligible DNA tailing, demonstrating no induction of significant double-strand DNA damage. These observations suggest that WX006 does not suppress tumor growth through structural DNA disruption but rather by impeding DNA synthesis, thereby slowing cell division (Appendix A).

Finally, we systematically evaluated the mechanism of WX006-induced cell death by employing pharmacological inhibitors targeting distinct cell death pathways, including Necrostatin-1 (necroptosis inhibitor), VX765 (pyroptosis inhibitor) and DIDS (chloride channel inhibitor). Notably, none of these inhibitors demonstrated significant attenuation of WX006-induced cell death in HCC cells. (Appendix A) In contrast, the cuproptosis inhibitor ammonium tetrathiomolybdate (ATTM) and the ferroptosis-specific inhibitor Ferrostatin-1 (Fer-1) both significantly reversed the cytotoxic effects of WX006 on Huh7 and Hep3B cells (Figure 1K,L). These results indicate that the anti-tumor efficacy of WX006 is primarily mediated through the induction of both ferroptosis and cuproptosis. We postulated that WX006 might disrupt intracellular iron and copper homeostasis, thereby triggering a synergistic lethal effect involving lipid peroxidation and mitochondrial copper-dependent proteotoxic stress. This novel hybrid cell death phenotype provides a clear direction for subsequent mechanistic investigations.

#### 3.2.3. Transcriptome Sequencing Revealed That WX006 Induces UPR and ISR in HCC Cells

Following the pharmacological validation of WX006, we performed high-throughput RNA sequencing (RNA-seq) to profile transcriptomic alterations in two HCC cell lines (Huh7 and Hep3B) before and after WX006 treatment, aiming to elucidate the compound’s mechanism of action. Significant transcriptomic alterations were observed in both cell lines following WX006 treatment. Notably, in Hep3B cells, the most prominently altered genes included *GADD45A/B/G*, *CHAC1*, *DDIT3*, *DDIT4*, *ATF3*, *WEE1*, and *ZFP36* (Figure 2A), while in Huh7 cells, the significantly changed genes were *DDIT4*, *CHAC1*, *IFITM1*, *IFIT2*, *OAS2*, *OAS3*, and *OASL* (Figure 2B).

Following the identification of differentially expressed genes (DEGs), pathway enrichment analyses, including KEGG (Kyoto Encyclopedia of Genes and Genomes) and Gene Ontology (GO) enrichment to identify significantly altered biological pathways and functional categories. Gene Ontology (GO) analysis across Biological Process, Cellular Component, and Molecular Function revealed that the DEGs in Hep3B cells were primarily enriched in terms related to chromatin organization, cell cycle progression, and transcription regulation (Figure 2C), whereas Huh7 cells showed protein folding and cellular component topology (Figure 2D).

We further performed Gene Set Enrichment Analyses (GSEA), which revealed distinct pathway enrichment patterns between Hep3B (Figure 2E) and Huh7 (Figure 2F) cell lines. In Hep3B cells, GSEA demonstrated significant enrichment in glucose starvation response, the endoplasmic reticulum (ER) unfolded protein response (UPR), cellular energy homeostasis, ER stress-induced intrinsic apoptosis, mitotic spindle organization, oxidative phosphorylation, mTORC1 signaling, and *ATF4*-mediated adaptation to ER stress. Conversely, Huh7 cells exhibited predominant enrichment in mitochondrial membrane disruption, E2F target genes, G2/M checkpoint regulation, glycolysis, hypoxia response, mitotic spindle dynamics, mTORc1 signaling, and the ER-UPR.

Experimental data indicate that WX006 treatment triggers Hep3B cells primarily engage pathways related to glucose deprivation, ER stress, and energy metabolism dysregulation pathways, while Huh7 cells predominantly exhibit alterations associated with mitochondrial dysfunction, G2/M checkpoint control, and glycolytic reprogramming. These pathway signatures align with prior transcriptomic profiles, cell cycle dynamics, and cell death phenotypes, collectively suggesting WX006 exerts antitumor efficacy through coordinated disruption of metabolic homeostasis and stress response mechanisms.

In Hep3B cells, WX006 triggers a metabolic crisis and exacerbates endoplasmic reticulum (ER) stress. The enrichment of glucose starvation response pathways suggests an impairment in glucose utilization, potentially through the inhibition of key glycolytic enzymes (e.g., *HK2*, *PKM2*), coupled with a suppression of mitochondrial oxidative phosphorylation (OXPHOS). This observed OXPHOS enrichment is consistent with compromised mitochondrial respiratory chain function. Concurrent inhibition of mTORC1 signaling likely represses energy-demanding anabolic processes, such as ribosome biogenesis, which aligns with the previously observed cell cycle arrest. The activation of the unfolded protein response (UPR), particularly the enrichment of *ATF4* signaling and the transcriptional upregulation of its targets *ATF3* and *CHOP*, indicates activation of the *PERK-ATF4-CHOP* axis, likely initiated by the accumulation of misfolded proteins in the ER lumen. Furthermore, the enrichment of mitotic spindle-associated genes points to chromosomal segregation defects. These defects may act synergistically with the replication fork collapse (as evidenced by DNA fiber assays) to exacerbate genomic instability.

In Huh7 cells, WX006 primarily induces mitochondrial dysfunction and glycolytic reprogramming. The signatures of mitochondrial membrane disruption suggest a collapse of the mitochondrial membrane potential or the opening of the mitochondrial permeability transition pore (mPTP), leading to reactive oxygen species (ROS) overproduction and cytochrome C release. The concomitant enrichment of glycolysis and hypoxia response pathways implies a compensatory upregulation of the Warburg effect following mitochondrial damage, which ultimately proves insufficient, culminating in metabolic collapse. This metabolic failure may subsequently promote ferroptosis, potentially amplified by mitochondrial ROS. The observed activation of the G2/M checkpoint, consistent with our cell cycle data, indicates a failure of checkpoint adaptation under sustained energy stress. The downregulation of E2F target genes corresponded with reduced E2F1 protein levels (as verified by Western blot), supporting the conclusion of a sustained cell cycle arrest and inhibited DNA synthesis. Furthermore, WX006-mediated modulation of mTORC1 signaling may exacerbate ER stress by suppressing autophagy, while the ensuing unfolded protein response (UPR) activation can, in turn, further inhibit mTORC1 activity. This establishes a vicious cycle that severely compromises the biosynthetic capacity and viability of the tumor cells.

To mechanistically validate the efficacy of WX006 in both HCC cell lines, we performed KEGG pathway enrichment and visualization analyses using Pathview. The analysis revealed a consistent and significant downregulation in the transcriptional levels of cyclin family genes in both Huh7 and Hep3B cells. Concurrently, we observed a marked activation of the *PERK-ATF4-CHOP* signaling axis within the endoplasmic reticulum protein processing pathway. Furthermore, our data indicated discernible activation in pathways related to ferroptosis, ER calcium signaling, and the *AMPK* signaling cascade, albeit to varying degrees. Notably, these compound-specific regulatory patterns were consistently reproducible across both cellular models, underscoring the robustness of WX006’s mechanism of action. (Figure 2G).

Collectively, our findings support a model wherein the antitumor efficacy of WX006 stems from its disruption of intracellular proteostasis, primarily through the induction of endoplasmic reticulum (ER) dysfunction, leading to widespread aberrant protein folding and assembly. Mechanistically, the downregulation of Bak/Bax-Calpain pathway components, coupled with the upregulation of *SEC61*, suggests that WX006-induced ER stress is driven by convergent insults from calcium ion dyshomeostasis, amino acid deprivation, and oxidative stress. Furthermore, the concomitant suppression of apoptotic pathway activation by WX006 provides a mechanistic rationale for the absence of significant apoptotic cell death observed in HCC cells, indicating a shift toward alternative cell death modalities.

Finally, using the STRING protein–protein interaction database and integrating multiple resources (Gene Ontology, KEGG, UniProt, WikiPathway, Reactome), we constructed a protein interaction network. Core networks were extracted using Cytoscape (v3.9.1) to model WX006’s mechanism of action in both HCC cell lines (Figure 2H,I). In Huh7 cells, WX006 primarily induced activation of interferon and chemokine signaling pathways, transcriptional dysregulation, UPR and ER stress via HSP and PERK, cell cycle arrest mediated by CDC20 and AURKA, and mitochondrial respiratory dysfunction. In Hep3B cells, WX006 similarly triggered UPR and ER stress; however, the growth inhibitory effects in Hep3B were strongly associated with the AP-1 transcription factor complex, a dependency not observed in Huh7 cells.

Notably, transcriptomic alterations induced by WX006 treatment differed between Hep3B and Huh7 cells. Specifically, Hep3B exhibited pronounced transcriptional changes in the *CHOP-ATF3* axis and *HRI-DELE1* signaling pathways. Considering the subtle differences in cell death modalities (as previously observed) and the distinct genetic backgrounds of these two cell lines, these findings suggest that while WX006 triggers a convergent ISR in both Hep3B and Huh7 cells, its mechanistic execution diverges. In Hep3B cells, activation of the CHOP pathway may drive apoptosis, whereas Huh7 cells likely employ alternative ISR effector mechanisms.

#### 3.2.4. WX006 Activates ER Stress and UPR Through p-eIF2α-ATF4 Signaling

Our previous transcriptomic enrichment analysis revealed that WX006 triggers ER stress and UPR via eIF2α-ATF4 signal activation, induces cell cycle arrest through CDC20 and AURKA dysregulation, and impairs mitochondrial respiration. Pharmacological intervention studies using calcium channel blocker 2-AD (Figure 3B), and intracellular calcium chelator BAPTA-AM (Figure 3C) and ER stress inhibitor 4-PBA (Figure 3D) demonstrated significant attenuation of WX006 cytotoxicity via CCK-8 assays, whereas PERK inhibitor ISRIB failed to rescue cell viability (Figure 3E). This paradoxical observation suggests calcium-mediated mitochondrial toxicity rather than canonical PERK-CHOP apoptosis drives cell death.

To validate these findings, we employed RT-qPCR analysis to investigate the transcriptional modulation of endoplasmic reticulum stress and calcium ion homeostasis-associated genes in WX006-treated Huh7 and Hep3B cells (Figure 3F). Experimental data demonstrated that WX006 significantly upregulated the expression of *ATP2A2* (sarco/endoplasmic reticulum Ca^2+^-ATPase), *CLCC1* (chloride channel CLIC-like 1), *ITPR1* (inositol 1,4,5-trisphosphate receptor type 1), *SEC61A1* (translocon-associated subunit alpha), *ATF6* (activating transcription factor 6), *EIF2AK3* (eukaryotic translation initiation factor 2 alpha kinase 3, PERK), and *ERN1* (endoplasmic reticulum to nucleus signaling 1, *IRE1α*). These coordinated transcriptional alterations provide mechanistic evidence supporting the hypothesis that WX006 induces synergistic cell death through multivalent activation of ER stress pathways coupled with dysregulation of calcium ion homeostasis. Furthermore, the observed upregulation pattern aligns with canonical UPR signaling cascades, confirming the compound’s capacity to elicit sustained endoplasmic reticulum proteotoxic stress in HCC models.

To further examine the influence of intracellular calcium alterations on tumor cell death, we performed calcium ion staining using Fluo4-AM and endoplasmic reticulum labeling with ER-specific fluorescent probes (Figure 3G). Results revealed that WX006 treatment markedly increased intracellular calcium levels (elevated Fluo4 fluorescence intensity), with calcium signals showing strong colocalization with endoplasmic reticulum markers. This calcium elevation was reversed upon IP3 receptor inhibition using 2-AD. Combined with the inhibitory effect of calcium channel blockers on cell death, these findings confirm the critical role of endoplasmic reticulum calcium release in WX006-induced lethality.

To assess the effects of WX006 on amino acid metabolism in tumor cells, this study analyzed transcriptomic sequencing data to determine the relative expression levels of amino acid transporters and key metabolic genes, while employing targeted mass spectrometry to quantify intracellular amino acid content in Huh7 cells. Experimental results demonstrated significant upregulation of *xCT* and *ASNS* expression in both Huh7 and Hep3B cells (Figure 3H). *PRODH* expression increased markedly in Huh7 cells but decreased in Hep3B cells, whereas *IDO1* expression was notably elevated in Huh7 cells (Figure 3I). Targeted mass spectrometry revealed substantial reductions in multiple amino acid concentrations within Huh7 cells (Figure 3J). These findings collectively indicate that WX006 reprograms amino acid metabolic networks, triggering nutrient stress and redox imbalance, ultimately leading to cell death. Supplementation with SAM or Met effectively attenuated WX006-induced cell death (Figure 3K), while other amino acids failed to reverse its growth-inhibitory effects (Appendix A). Integrated with prior observations of amino acid depletion, oxidative stress, and ferroptosis phenotypes, these results suggest that WX006 likely disrupts methionine metabolism—particularly one-carbon metabolism and trans-sulfuration pathways—compromising methylation homeostasis and redox balance. This metabolic perturbation impedes normal protein biosynthesis by impairing amino acid availability and energy supply, ultimately driving cell death.

#### 3.2.5. WX006 Depletes Intracellular NAD^+^ and NADP^+^ as Well as Disrupts One-Carbon Metabolism

To evaluate the impact of WX006 on the one-carbon metabolism in tumor cells, this study performed untargeted metabolomic profiling of Huh7 cells post-WX006 treatment. Key metabolites exhibiting significant accumulation included trimethylglycine, L-valine, sulcatone, 2-ketogulonolactone, and 4-hydroxyproline. Notably, 5,10-methylenetetrahydrofolate (5,10-CH_2_-THF), a pivotal intermediate in one-carbon metabolism, displayed marked accumulation (Figure 4A). As the core one-carbon donor for dTMP synthesis, the buildup of 5,10-CH_2_-THF suggests impaired downstream metabolism, potentially due to inhibition of methylenetetrahydrofolate reductase (MTHFR) or depletion of its cofactors (e.g., NAD^+^/NADP^+^). This metabolic blockade likely disrupts dTMP synthesis, leading to dUTP/dTTP imbalance, uracil misincorporation into DNA, and replication fork collapse—consistent with DNA-Fiber assay observations. Concurrent nucleotide synthesis defects and purine depletion may exacerbate ATP/GTP exhaustion, amplifying ISR. These findings collectively indicate that WX006 disrupts one-carbon metabolism, impairing nucleotide synthesis, methylation homeostasis, dNTP production, and redox balance, ultimately triggering metabolic collapse.

Pathway enrichment analysis of metabolomic data (Figure 4B and Appendix A) revealed significant perturbations in arginine/proline metabolism, core tumor carbon metabolism, D-amino acid metabolism, mTOR signaling, branched-chain amino acid metabolism, and ascorbate metabolism. These results highlight WX006’s broad metabolic reprogramming effects, centered on amino acid remodeling, energy stress, and proteostasis imbalance. Arginine metabolism via nitric oxide synthase (NOS) and polyamine synthesis pathways, along with proline-to-4-hydroxyproline conversion (mediated by *PRODH*), aligns with observed ROS generation and oxidative stress phenotypes.

Transcriptomic analysis demonstrated WX006-induced upregulation of *MTHFD2* and suppression of *ALDH1L1* in both Huh7 and Hep3B cells, with differential *ALDH1L2* regulation observed between cell lines. The concurrent accumulation of 5,10-CH_2_-THF and *MTHFD2* overexpression suggests impaired one-carbon flux (Figure 4C,D). Quantification of NAD^+^/NADH and NADP^+^/NADPH ratios via WST-8 assays revealed significant NADP^+^ depletion and reduced NADP^+^/NADPH ratios in both cell lines within 1–12 h post-treatment (Figure 4E), accompanied by NAD^+^/NADH ratio reduction (Figure 4F). Targeted metabolomics (Appendix A) confirmed suppression of glycolytic and TCA cycle intermediates, correlating with ATP depletion (Figure 4G).

Rescue experiments demonstrated that NAD^+^ (but not NADP^+^) supplementation attenuated WX006-induced cell death (Figure 4H and Appendix A), indicating NAD^+^ depletion plays a predominant role in mediating cytotoxicity. These results collectively establish that WX006 disrupts one-carbon metabolism through NAD^+^/NADP^+^ depletion, with NAD^+^ deficiency being the critical determinant of tumor cell lethality. The metabolic perturbations converge on nucleotide synthesis failure, redox imbalance, and energy crisis, driving irreversible metabolic collapse in HCC models.

#### 3.2.6. WX006 Induces Ferroptosis and Cuproptosis in HCC Cells

This investigation confirmed WX006-induced ferroptosis and cuproptosis in Huh7 and Hep3B tumor cells, given its previously identified hybrid cuproptosis/ferroptosis phenotype. RT-qPCR analysis demonstrated elevated expression of ferroptosis-associated genes, particularly *SLC7A11* and *CHAC1*, following WX006 treatment (Figure 5A). Subsequent co-treatment experiments with the iron chelator deferoxamine mesylate (DFOM) revealed no mitigation of WX006-induced cytotoxicity (Appendix A), while exogenous ferrous iron supplementation significantly attenuated cell death (Figure 5B). These paradoxical observations prompted two mechanistic hypotheses: WX006-induced oxidative stress might be counterbalanced by ferrous iron’s antioxidant properties, or WX006 disrupts iron-dependent metabolic processes through iron depletion or dyshomeostasis.

To differentiate these mechanisms, cells were co-treated with ROS scavengers (Trolox) and mitochondrial ROS inhibitors (mito-TEMPO). Neither intervention alleviated WX006-induced cytotoxicity, suggesting non-canonical ferroptosis mechanisms independent of ROS accumulation (Appendix A). Analysis of glutathione metabolism revealed progressive depletion of both GSH and GSSG pools in treated cells, accompanied by reduced GSH/GSSG ratios, indicating systemic collapse of antioxidant defenses rather than conventional redox imbalance (Figure 5C).

Quantitative colorimetric assays (Appendix A) and FerroOrange fluorescence imaging (Figure 5D) demonstrated significant intracellular ferrous iron accumulation post-treatment. Paradoxically, lipid peroxidation analysis via Liperfluo staining showed only modest increases, which were fully reversed by ferrostatin-1 (Fer-1) (Appendix A). Conversely, malondialdehyde (MDA) quantification revealed substantial lipid peroxidation at 6 h post-treatment, attenuated by multiple inhibitors targeting IP3R (2AD), PERK (ISRIB), autophagy (3MA/HCQ), copper chelation (ATTM), and mitochondrial ROS (mito-TEMPO). These findings position WX006-induced ferroptosis as a downstream consequence of integrated proteotoxic and oxidative stress cascades (Figure 5E).

Transmission electron microscopy revealed characteristic ultrastructural damage, including mitochondrial vacuolization and ER dilation within 3 h of treatment (Figure 5F). Functional assessments confirmed mitochondrial membrane potential collapse (JC-1 staining) (Figure 5G) and ATP depletion (Figure 5H), consistent with bioenergetic crisis. Copper quantification assays demonstrated time-dependent intracellular copper accumulation (Figure 5I). Notably, immunofluorescence detected dose-dependent DLAT oligomerization, a hallmark of copper-dependent cell death (Figure 5J).

Collectively, these data establish WX006 as a dual inducer of cuproptosis and ferroptosis through convergent mechanisms: Copper dyshomeostasis triggers DLAT oligomerization, disrupting mitochondrial metabolism, while Iron overload synergizes with glutathione system collapse to drive lipid peroxidation. The observed organelle-specific damage patterns (mitochondrial cristae loss, ER stress) and metabolic perturbations (ATP depletion, amino acid exhaustion) suggest coordinated targeting of multiple metal-dependent pathways, positioning WX006 as a novel polypharmacological agent against neoplastic cells.

#### 3.2.7. WX006 Disrupts Cellular Metal Homeostasis and NAD^+^ Metabolism in HCC Cells

This part of study investigated WX006’s disruption and metal ion dyshomeostasis in tumor cells respiratory chain function. UV spectral characterization demonstrated dose-dependent ferrous iron chelation capacity, with minimal copper-binding activity in vitro (Figure 6A and Appendix A).

Treatment with WX006 significantly inhibited all respiratory chain complex activities in isolated mitochondria from Huh7 and Hep3B cells. Ferrous iron supplementation and ATTM co-treatment effectively restored Complex I/Ⅱ functionality, while NAD^+^ supplementation showed partial recovery (Figure 6B,C). Complex Ⅲ/IV activities remained irreversibly impaired regardless of inhibitor combinations (Figure 6E). Subsequent ATP quantification demonstrated that ferrous iron supplementation substantially restored cellular ATP levels, with ATTM/NAD^+^ co-treatment showing moderate efficacy (Figure 6E,F).

Building upon prior findings that ferrous iron supplementation, copper chelation, or NAD^+^ restoration alleviates WX006-induced cytotoxicity, we systematically analyzed NAD^+^ metabolism under combinatorial treatments. Neither ER stress inhibitors nor calcium channel/autophagy modulators rescued NAD^+^ depletion, while ferrous iron or copper chelation restored cell viability without normalizing NAD^+^ metabolism. Notably, nicotinic acid/nicotinamide supplementation effectively reestablished NAD^+^ homeostasis, suggesting WX006-induced NAD^+^ depletion stems from excessive consumption rather than biosynthesis pathway inhibition (Figure 6G–I).

#### 3.2.8. WX006 Disrupts HCC Cellular Metabolism Through Binding to the NDUFS2 Site

Building upon previous findings, we hypothesized that beyond disrupting metal ion homeostasis, WX006 might exert structural inhibition on critical nodes within the NAD^+^ metabolic pathway. To investigate this, molecular docking and virtual screening were systematically conducted across key NAD^+^ metabolic targets. Computational analysis identified the ubiquinone (CoQ10) binding site of mitochondrial respiratory chain complex I as the most probable interaction locus. This site was subsequently designated as the binding pocket for structural investigations (Appendix A).

Using the DOCK6 platform, semi-flexible docking simulations generated a set of optimal binding conformations. Following energy minimization and scoring refinement, the predominant docking pose was subjected to detailed binding mode analysis (Appendix A). To validate computational reliability, the top-ranked conformation was further verified through independent docking simulations using the AutoDock Vina (Verision:1.2.5) platform, demonstrating consistent ligand-receptor interaction patterns across both docking methodologies. This multi-algorithm approach ensures robustness in predicting WX006’s potential binding mechanism at complex I.

Molecular docking studies identified a high-affinity interaction between WX006 and the CoQ10-binding pocket of Complex I (NDUFS2-NDUFS7-ND1, PDB:6ZTQ). DOCK6 analysis revealed strong binding (Grid Score = −69.33) mediated by hydrogen bonds with Tyr108/Gly61, π-π stacking with Phe86, and hydrophobic interactions with His59/Thr156/Ala73/Pro56/Leu159 (Appendix A, Figure 7A). AutoDock Vina validation confirmed stable binding (ΔG = −15.431 kJ/mol) (Appendix A).

To investigate the binding characteristics of WX006 within the NDUFS2-NDUFS7-ND1 pocket, we conducted 100 ns molecular dynamics (MD) simulations. The simulated complex exhibited excellent structural stability throughout the trajectory. Thermodynamic validation via MM-PBSA calculations yielded consistent binding free energies below −100 kJ/mol, robustly supporting the stability of the interaction (Figure 7B). Root mean square deviation (RMSD) analysis quantified atomic positional divergence from the initial conformation, revealing that the receptor pocket RMSD stabilized below 0.9 nm post-equilibration (Appendix A), confirming dynamic structural integrity. Further analysis demonstrated that WX006 maintained an average of two stable hydrogen bonds with the protein framework (Appendix A), while its ligand RMSD relative to the protein backbone remained at a low fluctuation level (~0.2 nm, Figure 7C and Appendix A). These MD simulations collectively confirm that WX006 forms a stable complex with the NDUFS2-NDUFS7-ND1 site of Complex I.

Given the predominant interaction between WX006 and NDUFS2 within the NDUFS2-NDUFS7-ND1 pocket, this study employed Drug Affinity Responsive Target Stability (DARTs) to validate their interaction (Figure 7D). Experimental conditions included a WX006 concentration gradient (0–1000 μM) followed by pronase digestion and subsequent quantification of NDUFS2 protein levels. Results demonstrated a dose-dependent stabilization of NDUFS2 protein, with enhanced resistance to proteolytic degradation correlating with increasing drug concentrations (Figure 7E,F). To further validate the DARTS findings, we implemented Cellular Thermal Shift Assay (CETSA) to assess WX006’s effect on NDUFS2 thermal stability. As shown in Figure 7G,H, WX006-treated NDUFS2 demonstrated a pronounced shift in thermal denaturation temperature (T_m_), increasing post-treatment (Figure 7I). This marked enhancement in thermal stability directly reflects WX006-NDUFS2 binding, which confers structural resistance to thermal denaturation. The CETSA results align with DARTs data, collectively confirming that WX006 directly engages and stabilizes the NDUFS2 receptor.

WX006 exerts its antitumor effects through two synergistic mechanisms in cancer cells. First, WX006 specifically binds to the CoQ10-binding site of mitochondrial respiratory chain Complex I, blocking the oxidation of NADH to NAD^+^ and inducing severe intracellular NAD^+^ depletion. This disruption impairs NAD^+^-dependent metabolic pathways, including glycolysis, the TCA cycle, one-carbon metabolism, and the glutathione redox system, leading to catastrophic energy metabolism collapse. Second, WX006 disrupts mitochondrial-cytoplasmic metal ion homeostasis via its metal-chelating properties. This interference compromises Fe-S cluster biogenesis, resulting in mitochondrial dysfunction characterized by loss of membrane potential (ΔΨ_m_) and excessive reactive oxygen species (ROS) production (e.g., hydrogen peroxide, H_2_O_2_) due to impaired electron transport. The accumulation of H_2_O_2_ triggers Fenton reactions with cytosolic Fe^2+^, generating hydroxyl radicals (•OH) that drive lipid peroxidation and oxidative damage. Concurrently, unresolved ISR and activation of the UPR exacerbate cellular injury. The glutathione-mediated antioxidant defense system is further suppressed by mitochondrial dysfunction and ER stress-induced upregulation of CHAC1, which degrades glutathione. These cascading effects collectively induce cuproptosis (via copper-dependent protein aggregation), ferroptosis (through iron-catalyzed lipid peroxidation), and bioenergetic failure, ultimately leading to cancer cell death.

#### 3.2.9. WX006 Demonstrates Safe and Effective Suppression of Hepatocellular Carcinoma (HCC) Growth In Vivo

Building on preliminary experimental evidence, we systematically investigated the therapeutic efficacy of WX006 in HCC-bearing animal models. (Figure 8A) WX006 demonstrated antitumor efficacy in both ectopic and orthotopic HCC models with favorable safety profiles. In H22 ectopic xenografts, one-week treatment induced significant tumor volume reduction (Figure 8B). Systemic safety was evidenced by stable body weight fluctuations (Figure 8C) and normal visceral indices across all dose groups (Figure 8D), with H&E staining revealing intact organ microarchitecture devoid of drug-induced lesions (Figure 8E).

In orthotopic Hepa1-6-luc models, in vivo fluorescence imaging quantified tumor burden reduction in treat group after 7-day treatment (Figure 8F,G), correlating with stable tumor weight (Figure 8H). Concurrently, preserved visceral indices and absence of histopathological abnormalities in major organs reinforced compound safety in both models (Appendix A). Immunohistochemical analysis showed dose-responsive suppression of Ki67 proliferation index (Figure 8I), confirming target engagement in hepatic tumors.

To further validate the clinical development potential of WX006 for HCC, we established an orthotopic HCC-bearing mouse model using Hepa1-6-Luc cells. Mice were orally administered WX006 or sorafenib at equivalent doses (50 mg/kg) for 5 consecutive days (Figure 8J and Appendix A). Bioluminescence imaging quantification revealed that WX006 exhibited superior anti-tumor efficacy compared to sorafenib (Figure 8K), while maintaining a favorable safety profile with no significant body weight loss or hematological toxicity observed (Figure 8L).

## 4. Discussion

This study presents the first report of WX006, a novel CAPE derivative featuring a unique molecular architecture, which exhibits remarkable metabolically driven tumor-selective cytotoxicity with minimal off-target effects on normal cells. This selectivity is closely associated with the abnormally elevated metabolic vulnerability of rapidly proliferating tumor cells [56]. WX006-mediated chelation preferentially disrupts iron-sulfur cluster biogenesis in tumor cells, concurrently impairing mitochondrial electron transport chain (ETC) integrity and crippling antioxidant defense systems. In contrast, normal cells effectively maintain metal ion homeostasis through metallothionein and glutathione buffering systems and demonstrate a superior capacity for repairing damaged iron-sulfur cluster proteins [57,58].

Tumor cells critically depend on mitochondrial Complex I (NADH dehydrogenase) activity to sustain NAD^+^/NADH balance and tricarboxylic acid (TCA) cycle functionality [59,60]. By competitively occupying the ubiquinone-binding pocket of Complex I, WX006 disrupts electron transfer kinetics, ultimately triggering a bioenergetic crisis (ATP depletion) and a redox catastrophe (excessive ROS production). Normal cells retain metabolic plasticity under conditions of Complex I impairment due to intact compensatory bioenergetic pathways, such as fatty acid β-oxidation, which sustain ATP production [61]. Furthermore, the activity of antioxidant systems, including superoxide dismutase (SOD), catalase (CAT), and glutathione (GSH), is significantly higher in normal cells compared to tumor cells [58]. Consequently, WX006-induced ROS can be rapidly cleared in normal cells, whereas they cause cumulative oxidative damage in tumor cells due to GSH depletion. This tumor-selective targeting provides a strong rationale for the clinical development of WX006, particularly for malignancies exhibiting mitochondrial or metal ion dependencies [62].

Emerging evidence suggests therapeutic potential in inducing crosstalk between ferroptosis and cuproptosis processes [63]. Concurrently, mitochondrial homeostasis has been identified as a crucial regulator of both ferroptosis and cuproptosis, establishing mitochondrial targeting as an emerging strategy in cancer therapeutics [64,65,66]. Current clinical approaches utilize various ferroptosis/cuproptosis inducers, such as RSL3, Erastin, and Elesclomol [63,67,68,69]. However, our findings reveal that WX006 induces cell death through mechanisms distinct from these classical pathways. It elicits cuproptosis via direct ETC disruption, with ferroptotic effects emerging secondarily to mitochondrial damage-induced ROS burst and antioxidant system failure. Rather than initiating a single cell death modality, WX006 triggers a hybrid pattern characterized by lipid peroxidation, ROS burst, copper-mediated proteotoxicity, mitochondrial dysfunction, ATP depletion, and endoplasmic reticulum (ER) stress, while lacking classical features of necrosis, pyroptosis, or apoptosis [70,71]. The subsequent molecular cascades, including ER stress and mitochondrial dysfunction, can amplify pathological alterations across multiple cellular components, ultimately enhancing immunogenicity and tumor cell death.

Targeting NAD^+^ metabolism presents a viable strategy for enhancing HCC therapy by modulating redox balance and overcoming chemoresistance [18]. Upregulation of the NAD^+^ salvage pathway, particularly via nicotinamide phosphoribosyltransferase (NAMPT), sustains HCC survival, and its inhibition by agents like FK866 depletes NAD^+^ pools, inducing mitochondrial dysfunction and ROS accumulation to sensitize tumors to sorafenib or cisplatin [72,73,74]. Several Complex I inhibitors have been explored [75], including rotenone, which induces apoptosis in tumor cells in preclinical models. Ongoing research is investigating the combination of NAD^+^ metabolism inhibitors with other therapeutic approaches, including chemotherapy and immunotherapy, to improve cancer treatment outcomes [76]. Accumulating evidence from multiple studies demonstrates that targeted modulation of mitochondrial function or strategic alteration of one-carbon metabolic phenotypes can effectively potentiate the efficacy of immunotherapeutic interventions against malignancies [77,78,79]. In contrast to classical mitochondrial disruptors like rotenone, WX006 exhibits a wider therapeutic window. Owing to its structural divergence from conventional Complex I inhibitors at the ubiquinone-binding interface, WX006 preferentially targets metabolically hyperactive tumor cells. This property enables the induction of mixed cell death modalities at lower therapeutic doses, achieving a broader therapeutic window and providing a compelling rationale for clinical translation.

Beyond the mechanisms described above, WX006 demonstrates substantial developmental potential in multiple areas. For instance, its combination with nano-scale drug delivery systems or chemotherapeutic/immunomodulatory agents may enhance therapeutic outcomes. Compared to widely reported nanometallic molecules [80,81], WX006 possesses the unique advantage of functioning without requiring exogenous metal ion supplementation or chelation [82]. Its therapeutic efficacy primarily stems from disrupting the metal equilibrium that supports cytoplasmic-mitochondrial metabolism, thereby initiating intracellular iron-copper-ER cascades that amplify mitochondrial pathology and effectively disrupt the efficient metabolic cycling in tumor cells [83].

## 5. Conclusions

Employing a structure-based optimization strategy focused on the catechol group, we synthesized a novel series of CAPE derivatives as potential prodrugs for hepatocellular carcinoma (HCC) treatment. The lead compound, WX006, demonstrated potent anti-HCC activity in Hep3B and Huh7 cell lines, coupled with lower toxicity in mouse primary hepatocytes, suggesting a wider therapeutic window than CAPE.

Mechanistic studies revealed that WX006 binds to the ubiquinone-binding site of NDUFS2, inhibiting Complex I and disrupting mitochondrial respiration. This, in combination with the disruption of intracellular metal ion homeostasis, alters redox balance and induces characteristics of ferroptosis and cuproptosis. Molecular dynamics simulations and MM-PBSA calculations over a 100 ns trajectory identified key residues stabilizing the WX006-NDUFS2 complex.

In summary, structural modification of CAPE yielded WX006, a derivative with enhanced antitumor potency and reduced toxicity, representing a promising chemotherapeutic candidate. While its undetermined pharmacokinetic profile currently limits clinical translation, further structural optimization, biological evaluation, and pharmacokinetic studies are actively underway.

Compound: The compound (WX006) used in this article was independently designed and synthesized by our research team and was used for subsequent pharmacological studies. It cannot be obtained through commercial channels.

## Data Availability

The High-throughput transcriptome sequencing data generated in this study are publicly available in NCBI Sequence Read Archive (SRA) at PRJNA1260226 and PRJNA1257854. All other data obtained and/or analyzed during the current study were available from the corresponding authors on reasonable request. Other raw data required to reproduce these findings cannot be shared at this time due to technical or time limitations. The processed data required to reproduce these findings are available to download from [Dong, Hao (2025), “Synthesis and Biological Evaluation of a Caffeic acid phenethyl ester Derivatives as Anti-Hepatocellular Carcinoma Agents via Inhibition of Mitochondrial Respiration and Disruption of Cellular Metabolism”, Mendeley Data, V1, doi: 10.17632/byp827tmcy.1].

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
