# Peer review of "Synthesis and Biological Evaluation of a Caffeic Acid Phenethyl Ester Derivatives as Anti-Hepatocellular Carcinoma Agents via Inhibition of Mitochondrial Respiration and Disruption of Cellular Metabolism"

_cancers, 2025, doi:10.3390/cancers18010092_

Round 1

Reviewer 1 Report

Comments and Suggestions for Authors

Attached, please find suggestions/corrections to the manuscript.

Author Response

All reviewer comments have been incorporated into the manuscript.Please see the attachment

Reviewer 2 Report

Comments and Suggestions for Authors

The manuscript by Dong et al studied an EAPE derivative’s anti-liver cancer effects using in vitro cell and in vivo mouse models. They experiment is well designed and data is comprehensive with the use of multiple methods. The study provides the chemical synthesis information, phenotype type observation, and mechanism studies.    

Here are some minor comments:

  1. On top of page 3, at the beginning of first paragraph, “Owing 40”, what does this mean? A reference? This and previous paragraphs need better transition and connection.
  2. In the third paragraph of page 3, it writes “by Group XX in 1988”. It seems like the information is missing.
  3. Only “scheme 3” is indicated in the text, but not “scheme 1” and “2”. What is the difference in structures between scheme 2 and 3? Which one is WH006? How about “Scheme 4” structures?
  4. How the IC50 is calculated in this study?
  5. All the figures are too small to read. Especially figures 2 and 4, it is so hard to read some of figures, labels, and words. The qualify of figures need to be improved.
  6. What is the rationale or evidence of anti-cancer effects of EAPE?

Author Response

  • On top of page 3, at the beginning of first paragraph, “Owing 40”, what does this mean? A reference? This and previous paragraphs need better transition and connection.

Response:

We have looked into this issue and believe the misunderstanding may have stemmed from the absence of certain details, which have now been included in the revised manuscript.

  • In the third paragraph of page 3, it writes “by Group XX in 1988”. It seems like the information is missing.

Response:

Upon reviewing this point, we believe the initial misunderstanding may have led to the absence of key details in our original submission. We have now clarified this by adding the relevant information in the revised manuscript.

  • Only “scheme 3” is indicated in the text, but not “scheme 1” and “2”. What is the difference in structures between scheme 2 and 3? Which one is WH006? How about “Scheme 4” structures?

Response:

In Scheme 2, we present several representative structures of CAPE derivatives. In contrast, Scheme 3 specifically focuses on CAPE derivatives that exhibit notable antitumor activity, distinguishing them from the broader anti-inflammatory or neuroprotective profiles outlined in Scheme 2. We considered it appropriate to group these antitumor-active derivatives separately here. Subsequently, Scheme 4 delineates the general synthetic route for WX006 and its structural analogues. All schemes, along with their respective distinctions and numbering, have been carefully reviewed and clearly marked in the revised manuscript.

  • How the IC50is calculated in this study?

Response:

In this study, the IC₅₀ values of the compounds were determined as follows: each compound was precisely weighed and dissolved in anhydrous DMSO to prepare stock solutions of 10 mM or 100 mM. During the assay, these stock solutions were freshly diluted to the desired concentrations with cell culture medium. An equal volume of anhydrous DMSO was diluted in the same manner and served as the vehicle control. Following treatment, cell viability was evaluated using the CCK‑8 assay, and absorbance was measured at 450 nm. The inhibition rate was calculated based on the absorbance changes. A dose–response curve was generated by plotting the inhibition rate on the y‑axis against log₁₀(drug concentration) on the x‑axis. The IC₅₀ values were then calculated using GraphPad Prism 9.5.3 through nonlinear regression (curve fit) with the “log(inhibitor) vs. response – variable slope (four parameters)” model.

  • All the figures are too small to read. Especially figures 2 and 4, it is so hard to read some of figures, labels, and words. The qualify of figures need to be improved.

Response:

We have adjusted the resolution of these images to ensure that reviewers can clearly discern the key information.

  • What is the rationale or evidence of anti-cancer effects of EAPE?

Response:

Based on literature review and preliminary evidence, CAPE has been confirmed to exhibit a range of biological effects, including anticancer activity. However, its precise anticancer mechanisms remain unclear. Most studies suggest that its antitumor effects may originate from its potent anti‑inflammatory and antioxidant properties, such as its notable inhibition of the NF‑κB signaling pathway. Nevertheless, its specific molecular targets have yet to be identified.

Although CAPE has indeed been shown to inhibit the proliferation of various tumor cell lines—with significant effects observed in certain cancer types—its efficacy remains limited in others, including hepatocellular carcinoma. Therefore, the development of CAPE derivatives and the improvement of its delivery systems represent a promising research direction. All of the above background is supported by relevant references in the Introduction section of the manuscript.

Reviewer 3 Report

Comments and Suggestions for Authors

Dear Author, 

The author reported on the Synthesis and Biological Evaluation of a Caffeic acid phenethyl ester Derivatives as Anti-Hepatocellular Carcinoma Agents via Inhibition of Mitochondrial Respiration and Disruption of Cellular Metabolism. In this manuscript, the Author Describes the design and synthesis of compounds. A total of 28 caffeic acid phenethyl ester (CAPE) derivatives were designed and synthesized. Author Identification of the lead compound. Among all derivatives, compound WX006 showed the strongest anti-proliferative effect. WX006 achieved IC₅₀ values of 3.332 μM and 3.764 μM after 48 hours—significantly lower than those of the parent compound CAPE. The author was chosen  WX006 for a deeper investigation into its antitumor efficacy and mechanisms of action. In the Mechanistic exploration approaches, they used High-throughput transcriptomics, metabolomics, and mitochondrial function analyses to elucidate its intracellular mechanisms. In vivo validation demonstrated that WX006 exhibited potent therapeutic efficacy in both H22 ectopic xenograft and Hepa1-6-Luc orthotopic xenograft murine models, achieving stronger tumor suppression than sorafenib while maintaining a favorable safety profile. Collectively, these results highlight the promise of CAPE derivatives—particularly WX006—as potential therapeutic candidates for hepatocellular carcinoma and provide a solid foundation for further pharmaceutical development.

Overall, this research on Caffeic acid phenethyl ester Derivatives a good, and the obtained results are also the best, for accepting the manuscript. Overall, the manuscript and research have novelty. I recommend acceptance after minor revision

Author Response

(1)Please write SAR and compare with WX006

Response:

Thanks for your insightful comment. Here are the SAR and comparison with WX006.

  1. Core Ring and Double Bond System
    The α,β-unsaturated conjugated double bond is essential for maintaining multiple biological activities. Its removal significantly reduces or abolishes activity.  Regarding phenolic hydroxyl groups, retaining the free catechol structure is crucial for antiproliferative activity.
  2. Ester Bond and Side Chain Modifications
    While retaining the ester bond as the core linkage, stereochemistry plays a significant role in modulating activity. For instance, introducing a methyl group at the α-position of the ester oxygen results in a notable stereochemical effect, where the R configuration demonstrates markedly superior antiproliferative activity compared to the S configuration.
  3. Effects of Phenyl Ring Substituents
    Introducing either electron-donating or electron-withdrawing groups on the phenyl ring does not lead to a significant enhancement of antiproliferative activity in CAPE analogs. This indicates that the electronic effects of phenyl ring substituents are not a key determinant of activity in the optimization of this series.
  4. Side Chain Length and Structural Extension
    Appropriate extension of the side chain alkyl length (up to C4) helps enhance antiproliferative activity, but excessive length (e.g., C5) leads to a decline. The most impactful optimization strategy is the construction of a di-side-chain structure, which exhibits significantly stronger activity than traditional mono-side-chain CAPE derivatives. The length of the ester linker (n=2–4) shows a positive correlation with anti-hepatocellular carcinoma activity.

Comparison with WX006
The high activity of WX006 fully corroborates the core conclusions of the SAR above. Firstly, its di-side-chain design is key to its significantly superior activity over the parent CAPE. Secondly, its activity trend aligns with the positive correlation of ester linker length within the optimal range. Finally, it retains the essential α,β-unsaturated double bond, which is the core pharmacophore. Notably, WX006 does not rely on complex phenyl ring substituent modifications, further indicating that the focus of optimization for this class of compounds should be on the rational expansion of the overall scaffold and the precise design of linkers, rather than local substitutions on the phenyl ring.

  • Have you taken mixed 1NMR for the 1H NMR Spectrum of 38a?Is there any difference in thedelta values for the S and R isomers

Response:

Thank you for your question regarding the 1H NMR spectrum of compound 38a. We actually recorded the NMR spectrum using a mixture of 38a and 38b, which corresponds to the racemate 38c. In the acquired spectrum, the chemical shifts (δ values) of the R-enantiomer (38a) and the S-enantiomer (38b) show minor but discernible differences, as expected for enantiomers in a chiral environment or under certain analytical conditions.

  • All the 28 examples have only aromatic ester group,why author not take the heterocyclicester group,if only simple ester group WX006 shows the better results

Response:

We sincerely thank the reviewer for this insightful suggestion. Indeed, during the early stages of this project, we attempted to introduce heterocyclic ester groups into the scaffold to further explore chemical space diversity. However, the synthetic routes for such structures proved to be challenging, with low yields in key steps and poor stability of the intermediates, which hindered the efficient and scalable preparation of sufficiently pure compounds for systematic derivatization and biological evaluation.

Given these practical synthetic constraints, we focused our current study on aromatic ester derivatives. Through careful optimization of side-chain structure, length, and substitution patterns, we successfully identified the lead compound WX006, which exhibits significantly enhanced activity. We fully acknowledge the potential pharmacological and structural benefits that heterocyclic esters could offer. Our team is currently working on improving the corresponding synthetic routes by optimizing reaction conditions, exploring new catalytic systems, and adjusting protection strategies, aiming to overcome the yield and stability limitations for future systematic evaluation of heterocyclic ester analogs.

We appreciate the reviewer's valuable suggestion, which provides important guidance for our ongoing research direction.

  • The author has done molecular docking(WX006 and the CoQ10-binding pocket of Complexl(NDUFS2-NDUFS7-ND1,PDB:6ZTQ)and molecular dynamics(MD)simulations.Both resultsvalidate with their results. Please make a separate section for each,similar to WX006 Induces Ferroptosis and Proptosis inHCC cells and other

Response:

Upon thorough review of the manuscript, we agree that your suggestion is highly pertinent and valuable. This revision contributes significantly not only to the clarity of the manuscript's structure but also to enhancing reader comprehension of our work.

Accordingly, we have now reorganized the manuscript to include a dedicated section for the molecular docking and target validation studies, as you recommended. We sincerely thank you for this constructive suggestion.

  • WX006 demonstrates safe and effective suppression of hepatocellular carcinoma(HCC)growthin vivo;the author has also taken WX006 and standard Sorafenib (50 mg/kg,oral)

Response:

Thank you for this comment. To ensure we address your point accurately, could you please provide a little more clarification or context regarding this specific concern? We want to be certain we fully understand your feedback before revising the manuscript.

  • The research Protocol is in very detail;if possible,make it short so it can be easily understoodby readers

Response:

Thank you for your suggestion. In our initial submission, we included the complete experimental protocols within the main manuscript to facilitate the review process for both the editors and reviewers. Taking your feedback into account, and to enhance the readability and conciseness of the manuscript, we have now moved the detailed protocols to the supplementary material, where they remain accessible for readers who wish to consult them.

7) I checked all the NMR.There are some minor mistakes in the number of H,please correct itbis((S)-1-phenylpropan-2-yl)3,3'-(4,5-dihydroxy-1,2-phenylene)(2E,2'E)-diacrylate (38b) Similarly,such mistakes of the other NMR data please check mistakes and correct them

Response:

Thanks for your careful review. We have corrected all of the mistakes in the number of H in the characterization of 1H NMR spectrum. For certain compounds with symmetric structures, the integrated proton count appearing as half of the total hydrogen number is a normal spectral phenomenon.
